# Healthcare Workers’ Knowledge about the Segregation Process of Infectious Medical Waste Management in a Hospital

**DOI:** 10.3390/healthcare12010094

**Published:** 2023-12-31

**Authors:** Andreas S. Miamiliotis, Michael A. Talias

**Affiliations:** Healthcare Management Postgraduate Program, Open University Cyprus, P.O. Box 12794, Nicosia 2252, Cyprus; andreas.miamiliotis@st.ouc.ac.cy

**Keywords:** infectious medical waste management, medical waste, hospital, colour-coding, segregation, public health, human health

## Abstract

Any hospital’s primary goal is to restore human health and save lives through health services provided to patients, but at the same time, hazardous wastes are produced. Inconsistent management of unsafe wastes might cause adverse effects and other issues for workers, the environment, and public health. Segregation is considered the critical stage in successful medical waste management. Mixing hazardous medical waste with non-hazardous medical waste will be avoided by correctly applying practices at the segregation stage. This study aimed to assess personnel’s knowledge about infectious medical waste and segregation practices used at six wards in Nicosia General Hospital. An analytical cross-sectional study was conducted, and data were collected through a structured self-administered questionnaire. The Statistical Package of Social Science (SPPS) version 25 was used with a minimum statistical significance of α = 0.05. The study population was nurses, nurse assistants, ward assistants, and cleaners working at the study wards. Out of 191 questionnaires, 82 were received, with a response rate of 42.93%. Most participants were female (72%) and nurses (85.4%). Participants had moderate knowledge about infectious medical waste management and good knowledge regarding segregation practices applied in their ward. Segregation was not carried out as it should have been, since most participants stated that infectious medical waste was mixed with non-hazardous medical waste. The number of correct answers the participants gave regarding the colour-coding of different medical waste categories was 67.5%, and only four answered correctly to all questions. Although participants knew segregation practices and the colour-coding process applied to medical waste, they did not use them satisfactorily. They applied methods regarding segregation without specific training, knowledge and guidance. Due to the issue’s importance, training programs must be implemented and performed.

## 1. Introduction

Through the years, changes in human activities and lifestyle have led to a rise in the generation of various types of waste [1], and the healthcare industry could not avoid being affected by these changes.

According to the World Health Organization (WHO), Medical Waste (MW) refers to all waste generated during human diagnosis, treatment or immunisation, mainly in healthcare facilities, research centres, and laboratories dealing with health procedures [2,3,4,5]. Hospitals are the most important producers of MW, since they provide health services 365 days a year, 24 h a day [2].

MW constitutes about 1–2% of waste generated within the community [6,7]. Based on the literature, MWs are divided into two major categories: Hazardous Medical Waste (HMW) and non-Hazardous Medical Waste (non-HMW). Around 85% of total MW is considered non-hazardous or general waste and does not cause any harm to human life and the environment. The remaining 15% is HMW, and due to its composition (infectious, toxic, radioactive), it can negatively affect human life and well-being, the environment, and the finances of healthcare facilities if not managed correctly [8,9,10,11]. As a result, it is considered the second most hazardous waste globally, after radiation waste [12]. According to the WHO, around 10% is Infectious Medical Waste (IMW), and the remaining 5% is toxic and radioactive waste [2,13]. The concern about the risk posed by mismanagement of HMW to human health and the environment increased dramatically in the 1980s, specifically in 1988, when syringes, needles and other HMW were found exposed on beaches on the east coast of the USA [14,15].

It is well known that effective Infectious Medical Waste Management (IMWM) protects public health and the environment if IMW is treated and disposed of safely and responsibly. Effective IMWM also helps to prevent the spread of infectious diseases like Hepatitis A and C and HIV (Human Immunodeficiency Viruses). Finally, proper IMWM helps reduce related costs associated with IMW treatment and disposal due to possible reductions in the quantity of IMW [16,17].

A clear definition of IMW is required for staff to correctly identify IMW from non-HMW. The purpose of IMW varies between countries, and its characterisation often depends on health professionals. In Cyprus, and according to the waste law of 2011 (Law 185(I)/2011) [18] and its amendments, IMW includes all MWs that are potentially infectious, such as waste contaminated with blood and other body fluids. These materials have been in contact with patients infected with highly contagious diseases and sharp objects like contaminated needles, syringes and surgical blades.

The amount of IMW depends on various factors. Some of them are

(1)the number of beds and the occupancy rate;(2)the size of the hospital and type of specialisation;(3)segregation procedures;(4)preference for single-use disposable equipment.

According to WHO, the IMW generated in high-income countries per occupied bed/day is higher than in low-income countries. However, in low-income countries, the actual amount of IMW produced is believed to be much higher since, in most cases, effective segregation procedures are not applied, and IMW is mixed with non-HMW [19]. Inconsistent segregation practices can be attributed to the staff’s lack of knowledge of correct segregation practices, regulations, and guidelines.

In all countries, most problems regarding the management of IMW are associated with non-compliance with directives, legal guidelines and regulations. Proper management, treatment and disposal of MWs are significant challenges in many developing and non-developed countries [20], primarily due to limited economic resources, leading to environmental and public health hazards. Although the procedures applied for IMWM differ from hospital to hospital, the problematic areas that need improvement are similar in almost all hospitals [21]. The IMWM process consists of a chain of procedures: segregation, collection, temporary storage, transportation inside and outside the hospital, final storage, treatment, and final disposal [22,23]. Segregation is considered the critical stage for successful MW management and minimising waste volume [24,25]. The segregation of MW means their distribution into various categories, specifically infectious, toxic, sharp objects, radioactive, and non-HMW [1]. Segregation is more effective if practised correctly at the point of generation [26]. In Nicosia General Hospital (N.G.H) and according to the colour-coding process, IMW is collected in yellow containers, boxes and bags; non- HMW in black bags; empty bottles and vials that previously contained medicinal/pharmaceutical substances in red boxes; and cytotoxic waste in purple containers. Factors associated with improper segregation include lack of knowledge, awareness, attitudes, practices and inadequate management and guidance [27].

The amounts of IMW increased globally in 2020 and 2021 when the COVID-19 pandemic was in full swing, since the COVID-19 virus involved infections by pathogenic microorganisms [28]. Almost all consumables such as surgical masks, gloves, sheets, drug/syringe wrappers, etc., as well as food leftovers from the patients mainly in the early stages of the pandemic, were placed under IMW umbrella, both for more safety and due to prevailing fear [29].

The COVID-19 virus first appeared in December 2019 in Wuhan, China [30]. It was declared a pandemic by the Director General of the WHO in March 2020, after more than 118,000 infections had been recorded in 114 countries [17,31]. The outbreak of the COVID-19 virus has created global concern and caused economic, social, and environmental impacts [32,33]. Singh et al. stated that at the pandemic’s peak, the city of Wuhan produced nearly 247 tons of medical waste daily, almost six times greater than the amount before the pandemic [34]. The COVID-19 virus had caused more than 697 million cases and nearly 7 million deaths worldwide by the end of October 2023 [35].

In Cyprus, IMW does not receive the required attention at the segregation stage, presenting significant deficiencies, with perhaps the most important being mixing IMW with non-HMW. In 2019, N.G.H. produced more than 268 tons of IMW, which is relatively high considering the available and occupied beds. 

Due to the COVID-19 pandemic, the quantity of IMW produced significantly increased by 18.45% in 2020, 75.90% in 2021, and 66.11% in 2022, compared to 2019 and before the pandemic. The quantities of IMW were affected mainly due to the increase in available beds, the growth of continuous use of personal protective equipment from staff and the addition of single-use equipment, practices implemented worldwide to prevent the transmission and spread of the virus. Vast quantities of IMW had been additionally created through the assessments for detecting the COVID-19 virus since they required special treatment in case of positive results. The pandemic, however, helped significantly spread awareness among those responsible and the public regarding the safe and effective management of IMW. In contrast, the effectiveness of their management systems helped to stop the spread of the COVID-19 virus [36].

The study objective was to assess personnel’s knowledge about IMW and segregation practices applied at six wards in N.G.H.

## 2. Materials and Methods

### 2.1. Study Settings

This study is an analytical cross-sectional conducted at six wards in N.G.H. from the beginning of June 2022 to the end of July 2022. Specifically, the analysis was employed in the Pathology A/Neurology ward, Pathology B/Neurology ward, General Surgery A and B wards and Orthopaedic A and Orthopaedic B/Urology ward. All six wards had 172 beds at capacity, of which 165 were available for inpatient treatment, representing 95.93%. All six departments had approximately 37% of the available beds in the General Hospital, not including daycare beds. 

### 2.2. Study Population

The study population included nurses, nurse assistants, ward assistants, and cleaners who frequently contact IMW. The study population was 191 workers from the wards above, regardless of age, gender, working experience and educational level. The study population included 151 nurses, 11 nurse assistants, 17 ward assistants, and 12 cleaners. From a total of 191 people, 82 questionnaires were collected, meaning a response rate of 43.92%. For various reasons, the rest of the workers chose not to answer the questionnaire, and their choice was respected entirely.

### 2.3. Sampling Technique

The method used for data collection was a survey questionnaire given to the study population. A validated, anonymous, structured, self-administered questionnaire was utilised for data collection. The administration of the final questionnaire was carried out by the researcher himself, with the help of the responsible Nursing Officers of the departments under study and members of the Special Committee for the Management of HMW at the Hospital. This approach was chosen due to the cyclical working hours of the participants and the difficulty of delivering the questionnaire to each participant. The questionnaire aimed to help researchers identify the level of personnel knowledge about IMW and segregation practices applied in the workplace. The questionnaire involved demographic and professional data, general questions for IMW, questions regarding applied segregation practices, training programs, and colour-coding of different MW categories at N.G.H. The questionnaire was pre-tested for reliability, and a pilot study was conducted to estimate the completion time, verify that the questions included were understandable and correctly expressed, and identify any possible barriers to data collection.

#### 2.3.1. Demographic Information

Parameters of the demographic data of the selected sample of 82 questionnaires included gender, age, education level, working position and status and work experience at N.G.H. and workplace ward.

#### 2.3.2. General Knowledge Questionnaire

This section consisted of 7 questions, each with two options, “yes” or “no”, with one correct answer. To analyse the results of the knowledge section of the questionnaire, each correct answer was scored with 1 point and each incorrect answer with 0 points. 

#### 2.3.3. Knowledge Segregation Choices

This section consisted of 7 questions, each with two options, “yes” or “no”, with one correct answer. Through these questions, the authors wanted to evaluate participants’ knowledge regarding segregation practices applied at their workplace ward, and assesses whether the participants knew what was being implemented in their workplace. To analyse the results concerning the segregation practices applied section of the questionnaire, if the average number of answers was closer to 1, the participants were aware of the methods used in their workplace. In comparison, if the average number of responses was closer to 2, they did not know.

#### 2.3.4. Training Programs Information

This questionnaire consisted of four questions, with “yes” or “no”. They refer to the issue of staff education and training in IMWM and whether it is believed that the training programs will be important in improving the staff’s knowledge and, at the same time, in optimising the IMWM procedures applied. To analyse the results of this section, a different approach was used to the previous two sections. More specifically, if the average of the responses was closer to 1, the participants’ choices were “yes”, while if the average responses were closer to 2, most were “no”. 

#### 2.3.5. Segregation and Colour-Coding of Different MW Categories

This section of the questionnaire consisted of six questions, each with four options, one correct and three incorrect answers. This section refers to the responses given by the participants regarding the colour-coding applied to the studied areas. The answers evaluated the participants’ knowledge regarding correct colour-coding, since each item of medical waste had to be discarded in a proper container, such as a container or bag.

### 2.4. Statistical Analysis

Statistical Package of Social Science (SPPS) version 25 was used for data statistical analysis. A minimum level of statistical significance was counted at α = 0.05, which has been established in the medical and social sciences field, and the *p*-value was rounded to 3 decimals. Hypotheses testing was carried out using *t*-test and ANOVA hypothesis procedures to check whether the average values between subcategories have a significant statistical difference. 

### 2.5. Code of Ethics

All relevant permissions were obtained before conducting the study. All questionnaires were anonymous, and confidentiality was ensured. Additionally, all participants were assured that no personal information would be shared by the Bottom of Form Protection of Natural Persons about the Processing of Personal Data and the Free Movement of such Data Law of 2018 (Law 125(I)/2018) [37].

## 3. Results

Table 1 shows the demographic and professional data of the selected sample of 82 questionnaires. The parameters included are gender, age, education level, working position and status, and work experience at N.G.H and the workplace ward.

Eighty-two questionnaires were received, with a response rate of 42.93%. Most participants were female 59 (72%), as opposed to males, of whom there were 23 (28%). The professional group that predominated in the sample were nurses, with a percentage of 85,4%, while ward assistants and cleaners responded with a rate of 7.3%. No nurse assistants completed the questionnaire. A rate of 69.5% of the participants had a bachelor’s degree, 18.3% held a master’s degree, and the remaining 12.2% had only completed high school education. Out of the total sample, 56.1% worked under permanent employment, 24.4% under temporary/indefinite jobs, and 19.5% on a contract basis. Most participants had work experience at N.G.H. and a ward equal to or less than five years. Out of 81 responses (1 participant did not mention the age), half had a period similar to or less than 35. 

Table 2 shows that most study participants (90.2%) knew that N.G.H. IMW was generated, while the remaining percentage (9.8%) did not. Almost half of the participants (54.9%), answered that all waste produced in N.G.H is dangerous. Around 2/3 of the participants knew the IMWM process, while the remaining 1/3 stated the opposite. 

Sixty-eight (68) of the respondents believed they play an essential role in managing IMW, while almost 70.7% of the participants believed there will be consequences if they do not work correctly with IMW. Most participants knew the measures to be taken if they came into contact with IMW. Out of 82 responses, only 26 participants stated that they were aware of the national legislation and regulations regarding the management of IMW, with a response rate of 31.7%. Based on the results of the questionnaires, we concluded that participants’ knowledge regarding IMWM was moderate, confirmed by the total average score of 4.85 out of 7.

Table 3 shows no statistically significant difference between the demographic and professional data examined and participants’ knowledge in IMW, since the *p*-value was higher than 0.05 in all cases. This study’s *p*-value of 0.05 or less is considered statistically significant (*p* < 0.05).

Table 4 deals with the knowledge of participants regarding knowledge segregation practices. All participants (100%) perceived that segregation of MW was necessary, but 11% (nine participants) could not identify the IMW produced in their workplace. Three-quarters of the participants stated that segregation of IMW from non-HMW was applied at the point of generation. The same percentage responded that they knew the colour-coding process, while 70 participants (85.4%) stated that it was applied. More than half of the participants (56.1%) believed that all staff applied the colour-coding process to segregate MW, but most of them (76.8%) stated that IMW was mixed with non-HMW. The remaining 23.2% considered that effective segregation was applied; each MW was placed in the correct container based on the colour-coding process and avoiding mixing IMW with non-HMW.

According to the results, participants’ knowledge of segregation practices was good (total average score 1.20).

Table 5 shows the significance between demographic and professional data examined and participants’ knowledge regarding segregation practices used.

The results show no statistically significant difference between the demographic and professional data examined and participants’ knowledge of applied segregation practices, since the *p*-value was higher than 0.05 in all cases.

Table 6 consisted of four questions, with the options “Yes” or “No”. They refer to the issue of staff education and training in IMWM and the importance that it is believed that the training programs will have in improving the staff’s knowledge and at the same time optimising the IMWM procedures applied.

Regarding training, the vast majority, 90.2%, stated that they were not trained, while 25.6% indicated that they had been trained at least once in the past. Regarding the topics included in the training, specifically legislation and regulations of IMWM, the answers were 52.4% and 47.6%, respectively. One of the most essential percentages recorded in this study was that 93.9% of participants indicated that training programs for IMWM would be really useful for their work, and would help them to perform their duties regarding IMWM more efficiently. 

Table 7 shows the significance between demographic and professional data examined and participants’ answers regarding training programs.

Regarding participants’ responsiveness to training programs (total average score 1.55), it appears that staff lacked training programs, even though they believed that training programs benefit their work. Additionally, based on the results, there was no statistically significant difference between the demographic and professional data examined and training programs, since the *p*-value was higher than 0.05 in all cases. 

Table 8 refers to the responses given by the participants regarding the colour-coding applied to the studied sections. The answers in Table 8 evaluated the participant’s knowledge regarding the correct use of the colour-coding process, since each MW had to be discarded in the right container, such as a container or bag.

A response rate of 80.5% was found for correct answers regarding the segregation of gauze and gloves used on patients with infectious diseases. Additionally, correct answers regarding human tissues and blood-contaminated waste (considered IMW) were 65.9% and 92.7%, respectively. Regarding the segregation of empty ampoules that previously contained pharmaceutical substances, only 9.8% responded correctly. The remaining 90.2% stated the opposite. Response rates of 89% and 67.1% were found for correct answers regarding the segregation of outside packaging of medical consumables and pharmaceutical products and intravenous fluids in plastic containers, respectively. Correct answers were given by the participants in 67.5% of cases, corresponding to 332 correct answers out of 492. 

## 4. Discussion

Based on the questionnaire results, we concluded that participants’ knowledge of IMWM was moderate, confirmed by the average score of 4.85 out of 7. Additionally, there was no statistically significant difference between the demographic and professional data examined and participants’ knowledge of IMW, since the *p*-value was higher than 0.05 in all cases. This study’s *p*-value of 0.05 or less is considered statistically significant (*p* < 0.05). The findings and their implications should be discussed in the broadest context possible. Future research directions may also be highlighted. The most significant proportion of the sample (72%) was females, unlike males (28%). The predominance of the female gender and female nurses was expected, since the number of female workers and female nurses in N.G.H. is more significant, as this stands for most hospitals worldwide. Findings regarding the predominance of female nurses agreed with the data prevailing in the USA for 2008 [38], where male nurses were the minority in healthcare organisations. The findings also agreed with the study of Musa et al., where 91.3% of the participants were female nurses [39].

Almost nine out of ten participants had a bachelor’s or master’s degree. The result may be affected by the fact that most participants were nurses, and according to Parliament’s decision in 2007, nursing education was upgraded to university level. All nurses in Cyprus were then given the opportunity to upgrade their diploma to a bachelor’s degree.

Most respondents considered all MW generated to be hazardous, so there is a big possibility that all MW was collected in yellow containers, creating an unnecessary increase in IMW and hospital expenses. IMW treatment costs around 10–20 times more than non-HWM management. Therefore, adequate segregation is a significant financial factor for any hospital, since the cost of the treatment of IMW [40] and the risk of infection, mainly through injuries, will probably be reduced. A similar result was also reported in the study by Deress et al., where more than half of nurses stated that all MW was dangerous [41]. 

It is not only essential for staff to know the IMWM process applied in their workplace; it is also their obligation. Also, the administration must inform all staff about IMWM processes. Based on participants’ answers, nearly 1/3 were not assured of the IMWM process, so they probably did not follow correct practices. For example, the fact that 54.9% considered all MW produced in N.G.H. hazardous, and only 65.9% knew the IMWM process were elements that needed to be corrected immediately by the N.G.H. administration. The result was similar to that reported in the study of Deress et al. concerning nurse participants [41]. 

The percentage of 31.7% who stated that they were aware the national legislation and regulations regarding IMWM was deficient, which indicates that the hospital’s management and staff may not have paid the required attention to IMW management. This percentage was considered unsatisfactory, since only three out of ten participants were aware of the legislation in an area that is important for the performance of their work, their safety, the safety of patients and visitors, the wider public and the environment, and also for the hospital’s budget. These findings agreed with the study by Sharma et al., wherein it was reported that 31.7% of nurses were sufficiently aware of the regulations and legislation governing IMWM [42]. In contrast, Mathur et al. said that the percentage of nurses knowledgeable about laws and legislation was 91.7%, much higher than that recorded in this study [43]. 

The effectiveness of IMW procedures regarding segregation implies essential requirements, Such as the ability of the responsible staff to recognise the various types of MW and the availability of all necessary consumables. Many researchers consider the segregation stage the most critical stage of infectious waste management; when implemented correctly, the success of the control is almost guaranteed. The WHO recognises the importance of the segregation stage, and this was also recognised by all participants. This result was substantially better than the findings of Romin and Pensiri’s study, wherein 86.8% believed that the segregation of MW is essential [44]. 

Ideally, supposing the colour-coding process is executed correctly, the mixture of different types of MW will be prevented, resulting in a positive effect on the environment, public health, and the hospital itself. A rate of 85.4% believed that segregation of MW and the colour-coding process were applied, although only 75.6% knew of the colour-coding process used in their workplace. Based on these results, we can presume that some participants might not know the proper colour-coding process. Similar results were concluded in the study of Deress et al., 2018, where 73.6% of nurses responded that they were aware of the colour-coding process [41], while in the study of Maluni et al., 2018, 32% of the respondents were not aware of the segregation process [45]. 

Based on the responsiveness, segregation was not applied as it should have been, and a mixture of non- HMW with IMW occurred. Inadequate segregation may lead to an increase in hospital expenditure, since the cost of treatment of IMW is much higher than that of non-hazardous trash. N.G.H. compensates EUR 1.45 per kg for IMW, which includes collection from the final storage room, transport, treatment, and final disposal. On the other hand, compensation for the same services for managing non-hazardous waste is EUR 100 per route, transporting 1.5 tons per route on average. The results were in agreement with the study in Tanzania by Manyele and Lyasenga, from 2010, wherein it was reported that segregation procedures for IMW were applied, but not at a satisfactory level, which required improvement [46]. 

A necessary procedure is the segregation of MW at the point of generation, a practice encouraged and supported by WHO. If segregation of IMW is applied at the end of age, a mixture of different types of MW and injuries and transmission of infectious diseases will be avoided [47]. All healthcare professionals must know that IMW should never be mixed with non- HMW. Still, different types of HMW should not be combined. Segregating IMW at the point of production is the cornerstone for their proper management. At the same time, practices such as minimisation/reduction, reuse and recycling, reuse and recycling should be applied where possible [48]. In collaboration with the supervisors, the administration of N.G.H must promote the segregation of IMW at the point of generation, since 1 in 4 participants believed it was not applied. Deress et al. 2018, report in their study that 84% of nurses believe that segregation of IMW should be used at the point of generation [41]. 

Table 8 demonstrates more accurate results for participants’ knowledge of segregation and colour-coding processes. Participants’ answers were incorrect at a rate of 32.5%, meaning that participants chose the wrong colour. The result was unsatisfactory, since 1/3 of MW was not placed in the correct container, causing a mixture of various types of MW. As mentioned before, an inadequate segregation process may cause adverse effects on the public and the environment and additional hospital costs. As clearly stated in the results, the segregation and colour-coding process demands significant improvement. Notably, only four participants (less than 6%) responded correctly to all six questions regarding the segregation of MW based on colour-coding of different MW categories. Results reinforce from most participants that IMW was mixed with non- HMW, and all staff did not apply the colour-coding process. Responses of participants regarding the segregation of MW based on the colour-coding process also prove the lack of sufficient knowledge in the team, and the need for regular training programs. Similar results were found in the study by Maluni et al., where more than 20% of the participants placed MW in the wrong containers [45]. 

Regarding training programs, our results showed that most participants were never trained in IMWM. This result agreed with the findings of the study by Ali et al., 2015, which stated that educational programs were not implemented [49]. The percentage of participants who responded that they were trained in IMWM at least once in the past was almost similar to the 31% reported in the study conducted in Ethiopia by Hayleeyesus and Cherinete, 2016 [50]. However, it was lower than the 36.8% and 61.6% reported in the survey by Deress et al., 2018 and Uddin et al., 2014, respectively [41,51]. 

From participants’ replies, it was concluded that most support the idea that training programs should be carried out regularly to help them perform their tasks correctly and confidently. It is almost certain that participation in training programs improves both knowledge and efficiency of procedures. The result is similar to that of the study by Shivalli and Sanklapur 2014, where almost nine out of ten nurses (86%) expressed the need for continued training [24]. 

Results concerning training programs were a direct message to the administration of N.G.H. that staff were optimistic about being trained regularly and updating their knowledge regarding IMWM. 

Any administration should not ignore the importance of continuous education and training in IMWM in any hospital, since the human factor is as important as the technology, if not more so [52]. Training programs must be carried out in all professional groups and adapted according to the nature of the work and the educational level and the actual needs of each professional group. Staff training programs must include all parameters related to managing IMW. Specifically, an analysis of the following factors should be performed: Legislation on IMWM issues;The responsibilities of each employee, regardless of professional group;The guidelines for implementing good management practices of IMW, such as colour-coding;Practices regarding preventing/minimising IMW production;The risks related to managing IMW and the effects on human health and the environment;The importance of using personal protective equipment;Essential reporting procedures for each accident response (injury, spillage).

All operators must know how to behave and protect themselves, and systematic training and frequent information are perhaps the most important measures. The proper and effective management of IMW is a challenging task; it requires sufficient knowledge, teamwork, and cooperation, since one functionary must support and complement the other. It is the responsibility and obligation of all operators, regardless of their profession, their earnings, or their educational level, to correctly apply the procedures for the effective and safe segregation of MW; it also their responsibility to apply all the guidelines for the secure management of all types of MW, and especially of HMW.

Therefore, training programs should be included in the hospital’s policies and procedures. Adopting training programs for all newly hired staff and, more particularly, for newly hired nurses would also be a positive approach, as they would then know from the beginning how to act and behave in matters of IMWM and, more particularly, segregation procedures.

Many studies have concluded that one of the most critical factors to improve the IMWM process in healthcare facilities is upgrading knowledge levels, which is achieved primarily through systematic training [53,54,55].

Hospital administration should provide regular education and training to their staff on proper IMWM, and inform them about the latest regulations and practices. Furthermore, hospital administration must ensure that all personnel are familiar with the hospital’s policies and procedures regarding segregation.

This study also has some limitations. The participants probably answered the questionnaire with some bias according to their desires and interests, without impartiality. In addition, the sample in some cases could have been more significant, in order to derive more statistical test results. This study could be repeated with a larger sample in the future.

## 5. Conclusions

Based on present results, staff require regular training programs regarding IMWM, primarily on segregation practices. All staff should be obligated to identify all types of MW generated in their workplace and apply relative segregation practices. Moreover, the team should have the knowledge required to distinguish HMW from non- HMW, and place them according to the colour-coding process; they should know when practices applied are correct and when they are not.

Since IMWM plays a vital role in hospital management, public health and the environment, a concerted effort is required by all personnel. By empowering personnel’s knowledge and practices regarding segregation, the production of IMW could be minimised, and expenses for their treatment could be decreased.

An important aspect that needs improvement is the contribution of the supervisors of each ward in terms of MW segregation practices, through regular and intensive audits, to guide staff properly. Regular audits and inspections can help identify areas for improvement and ensure compliance with requirements. Due to the seriousness and importance of the issue, it is strongly suggested that training programs be held for all hospital staff.

## Figures and Tables

**Table 1 healthcare-12-00094-t001:** Questionnaire data of demographic characteristics.

Questionnaire Attributes	Categories	Count	Column N %
Gender	Male	23	28.0%
	Female	59	72.0%
Education level	High school	10	12.2%
	Bachelor	57	69.5%
	Master	15	18.3%
Working position	Nurse	70	85.4%
	Ward assistant	6	7.3%
	Cleaner	6	7.3%
Working status	Permanent	46	56.1%
	Temporary/indefinite	20	24.4%
	On contract basis	16	19.5%
Age of respondents	<=35	41	50%
	36–50	27	32.9%
	51+	13	15.8%
	No answers	1	1.3%
Work experience at N.G.H	<=5	33	40.2%
	6–15	29	35.4%
	16+	20	24.4%
Work experience at workplace (ward)	<=5	41	50.0%
	6–10	17	20.7%
	11+	24	29.3%

**Table 2 healthcare-12-00094-t002:** Frequency of study participants regarding general knowledge among each item question.

Questions	Answers	
	Yes	No
I am aware that N.G.H. is generating infectious medical waste.	74 (90.2%)	8 (9.8%)
All medical waste generated in N.G.H. is hazardous.	45 (54.9%)	37 (45.1%)
I am aware of the infectious medical waste management process in N.G.H.	54 (65.9%)	28 (34.1%)
I have an essential role in the infectious medical waste management process.	68 (82.9%)	14 (17.1%)
There are consequences for my work if I do not manage infectious medical waste correctly.	58 (70.7%)	24 (29.3%)
I am aware of the safety measures I must take whenever I come into contact with infectious medical waste.	74 (90.2%)	8 (9.8%)
I know the national legislation and regulations regarding managing infectious medical waste.	26 (31.7%)	56 (68.3%)
**Total average score**	4.85	

**Table 3 healthcare-12-00094-t003:** Significance of general knowledge in IMW and demographic and professional data.

Variable	Categories	Mean Values	*p*-Value
Gender	Male	4.87	0.953 (*t*-test)
	Female	4.85	
Age of responses	<=35	4.95	0.162 (ANOVA)
	36–50 51+	4.44 5.38	
Educational level	High school	4.70	0.795 (ANOVA)
	Bachelor	4.93	
	Master	4.67	
Working position	Nurse	4.94	0.140 ANOVA)
	Ward assistant Cleaner	3.67 5.00	
	Permanent	4.80	0.801(ANOVA)
Working status	Temporary/Indefinite On contract basis	5.05 4.75	
Work experience at workplace (ward)	<=5 6–10 11+	4.80 4.82 4.96	0.924 (ANOVA)
Work experience at N.G.H	<=5 6–15 16+	4.85 4.83 4.90	0.987 (ANOVA)

**Table 4 healthcare-12-00094-t004:** Frequency of study participants’ knowledge of segregation practices applied.

Questions	Answers	
	Yes	No
Segregation of medical waste is necessary.	82 (100%)	0 (0%)
I recognise the infectious medical waste produced at my workplace.	73 (89%)	9 (11%)
Segregation of infectious medical waste from non-hazardous medical waste occurs at the point of generation.	62 (75.6%)	20 (24.4%)
Medical waste is segregated according to the colour-coding process.	70 (85.4%)	12 (14.6%)
I am aware of the colour-coding process applied regarding medical waste.	62 (75.6%)	20 (24.4%)
All staff apply the colour-coding process for the segregation of medical waste.	46 (56.1%)	36 (43.9%)
There is a mix of infectious medical waste with non-hazardous medical waste.	63 (76.8%)	19 (23.2%)
**Total average score**	1.20	

**Table 5 healthcare-12-00094-t005:** Significance between segregation practices and demographic and professional data.

Variable	Categories	Mean Values	*p*-Value
Gender	Male	1.21	0.783 (*t*-test)
	Female	1.20	
Age of responses	<=35	1.17	0.307 (ANOVA)
	36–50 51+	1.24 1.22	
Educational level	High school	1.24	0.449 (ANOVA)
	Bachelor Master	1.21 1.15	
Working status	Permanent	1.24	0.077 (ANOVA)
	Temporary/Indefinite On contract basis	1.13 1.19	
Work experience at workplace (ward)	<=5 6–10 11+	1.17 1.24 1.23	0.231 (ANOVA)
Work experience at N.G.H	<=5 6–15 16+	1.16 1.24 1.23	0.102 (ANOVA)

**Table 6 healthcare-12-00094-t006:** Frequency of study participants’ responses regarding training programs.

Questions	Answers	
	Yes	No
I am training every year in infectious medical waste management issues.	8 (9.8%)	74 (90.2%)
I have been trained in infectious medical waste management issues in the past.	21 (25.6%)	61 (74.4%)
Both legislation and regulations regarding the management of infectious medical waste are included in the training programs of the N.G.H.	43 (52.4%)	39 (47.6%)
Infectious medical waste management training is very useful for my work.	77 (93.9%)	5 (6.1%)
**Total average score**	1.55	

**Table 7 healthcare-12-00094-t007:** Significance between training programs and demographic and professional data.

Variable	Categories	Mean Values	*p*-Value
Gender	Male	1.57	0.665 (*t*-test)
	Female	1.54	
Age	<=35	1.52	0.358 (Anova)
	36–50 51+	1.54 1.63	
Educational level	High school	1.60	0.623 (Anova)
	Bachelor	1.55	
	Master	1.50	
Working status	Permanent	1.59	0.070 (Anova)
	Temporary/Indefinite On contract basis	1.44 1.55	
Work experience at workplace (ward)	<=5 6–10 16+	1.53 1.54 1.57	0.810 (Anova)
Work experience at N.G.H	<=5 6–15 16+	1.52 1.53 1.60	0.538 (Anova)

**Table 8 healthcare-12-00094-t008:** Segregation and colour-coding of different HCV categories.

Questions	Answers	
	Right	Wrong
Gauzes and gloves used on patients with infectious diseases	66 (80.5%)	16 (19.5%)
Empty ampoules that previously contained pharmaceutical substances	8 (9.8%)	74 (90.2%)

Outside packaging of medical consumables and pharmaceutical products	73 (89%)	9 (11%)

Blood-contaminated waste	76 (92.7%)	6 (7.3%)
Human tissues	54 (65.9%)	28 (34.1%)
Intravenous fluids plastic container	55 (67.1%)	27 (32.9%)
Percentage of correct answers	67.5%	

## Data Availability

The datasets used and analysed during the current study are available from the corresponding author upon reasonable request.

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
