# Peer review of "Healthcare Workers’ Knowledge about the Segregation Process of Infectious Medical Waste Management in a Hospital"

_healthcare, 2023, doi:10.3390/healthcare12010094_

Round 1

Reviewer 1 Report

Comments and Suggestions for Authors

This valuable paper examines healthcare workers' awareness of infectious waste.

1. Please specify in the text where the explanation of Tables 4 to 9 can be found.

2. Table 6: When you say 'Infectious waste management training is really very useful for my work', what specific training do you envisage?

3. The answers seem to be extremely biased towards either "Right" or "Wrong". Is there a reason for this?

4. The need for training is well understood. However, I would expect you to be more specific about what information you are currently basing your work on and what specific training is required.

Author Response

This valuable paper examines healthcare workers' awareness of infectious waste.
Thank you for your positive attitude.
********************************************************************************************************
1. Please specify in the text where the explanation of Tables 4 to 9 can be found.
Ans 1: We have done it.
******************************************************************************************************
2. Table 6: When you say 'Infectious waste management training is really very useful for my work',
what specific training do you envisage?
Ans 2. We answered the lines 417 -449
****************************************************************************************************************
*
3. The answers seem to be extremely biased towards either. Is there a reason for this?
Ans 3
Thank you for your valuable comment. This is a binary variable. The answer is either "Right" or
"Wrong” by definition. This is so because in Table 8, we have six questions to check the
knowledge of healthcare workers and whether they know the correct answer. We do not use a
Likert scale because the possible answers are "Right" or "Wrong." Likewise for other tables
************************************************************************************
4. The need for training is well understood. However, I would expect you to be more specific about
what information you are currently basing your work on and what specific training is required.
Ans see the comments attached to the tables Page 8 ‐10 and the discussion

Reviewer 2 Report

Comments and Suggestions for Authors

Thank you for sharing this manuscript with me. The manuscript provides a good overview of healthcare workers' knowledge of infectious medical waste segregation practices in a hospital setting. The topic is relevant and important for improving healthcare waste management. Here are some constructive comments and suggestions:

l   The description of the study subjects needs to be more detailed, including why six wards in N.G.H. were chosen, the distribution of job positions among the 191 workers, and whether the comparison with the actual target population is representative.

l   Briefly describe how the questionnaires were distributed to the participants.

l   Add a subsection to describe 'Study measures.' Describe the process of questionnaire development in more detail. Mention how the questionnaire items were formulated, whether they were based on existing scales or guidelines, and how you ensured the questionnaire's reliability and validity.

l   Statistical Analysis: mention any specific statistical tests used, especially if they are relevant to the research questions.

l   Provide a discussion of the response rate (42.93%). Consider addressing the potential implications of this response rate on the study's results and conclusions.

l   Because the Materials and Methods do not describe the content of the questionnaire, measurement scales, and scoring methods, it is difficult for readers to understand the significance of the author's statistical results.

l   According to journal writing conventions, tables presenting research results should be placed after the main text. This manuscript is the opposite. It is recommended that the authors make appropriate modifications.

l   In Table 2, there are several items where the sum of 'yes' and 'no' exceeds 100%, indicating an error. Additionally, it's not clear if 'yes' and 'no' for each item represent the respondents' answers or correct/incorrect answers. The authors should label 'yes' and 'no' for each item and clarify the precise meaning of the numbers in the table. The suggestions in the latter part also apply to Table 4.

l   Tables 3, 5, and 7 should present the statistical mean values for each subcategory and the methods of analysis used.

l   Table 9 was not labeled in the text, and it can be integrated into Table 8.

l   The statement, 'In this study, a p-value of 0.05 or less is considered to be statistically significant (p<0.05). The findings and their implications should be discussed in the broadest context possible,' is unnecessary.

l   In this manuscript, discussing the gender and education level of nursing staff is not meaningful and is unrelated to the research topic.

l   The authors should specify the specific training content based on the results of this study in the training program.

l   Acknowledge any limitations of your study, such as the sample size or potential biases, to maintain transparency and reliability.

Author Response

Reviewer 2

Thank you for sharing this manuscript with me. The manuscript provides a good overview of healthcare workers' knowledge of infectious medical waste segregation practices in a hospital setting. The topic is relevant and important for improving healthcare waste management. Here are some constructive comments and suggestions:

Answer: Thank you for your positive words. Below we provide answers to each comment you raised

************************************************************************************************************

The description of the study subjects needs to be more detailed, including why six wards in N.G.H. were chosen, the distribution of job positions among the 191 workers, and whether the comparison with the actual target population is representative.

Answer: Thank you. We gave an answer 122 - 165

*******************************************************************************************************************

Briefly describe how the questionnaires were distributed to the participants.

Answer: We replied  in lines 143 - 165

************************************************************************************************

Add a subsection to describe 'Study measures.' Describe the process of questionnaire development in more detail. Mention how the questionnaire items were formulated, whether they were based on existing scales or guidelines and how you ensured the questionnaire's reliability and validity.

Answer: There is no issue of validity. We do not use latent variables and it is not a psychometric

****************************************************************************************************************

Statistical Analysis: mention any specific statistical tests used, especially if they are relevant to the research questions.

Answer: Done it.

*************************************************************************************************************

Provide a discussion of the response rate (42.93%).Consider addressing the potential implications of this response rate on the study's results and conclusions.

Answer: We mentioned the limitations of the study  in lines 459 - 462

***************************************************************************************************************

Because the Materials and Methods do not describe the content of the questionnaire, measurement scales, and scoring methods, it is difficult for readers to understand the significance of the author's statistical results.

Answer: We mentioned all the relevant details in the discussion of the tables results

***********************************************************************************************************

According to journal writing conventions, tables presenting research results should be placed after the main text. This manuscript is the opposite. It is recommended hat the authors make appropriate modifications.

Answer: We have done it.

**********************************************************************************************************

In Table 2, there are several items where the sum of 'yes' and 'no' exceeds 100%, indicating an error. Additionally, it's not clear if 'yes' and 'no' for each item represent the respondents' answers or correct/incorrect answers. The authors should label 'yes' and 'no' for each item and clarify the precise meaning of the numbers in the table. The suggestions in the latter part also apply to Table 4.

Answer: We have done it. Explain the philosophy of tables and the results on it

***********************************************************************************************************

Tables 3, 5, and 7 should present the statistical mean values for each subcategory and the methods of analysis used.

Answer: We have done it

************************************************************************************************

Table 9 was not labeled in the text, and it can be integrated into Table 8.

Answer: We have done it

*******************************************************************************

The statement, 'In this study, a p-value of 0.05 or less is considered to be statistically significant (p<0.05). The findings and their implications should be discussed in the broadest context possible,' is unnecessary.

Answer: We have removed the unnecessary comments.

**************************************************************************************

In this manuscript, discussing the gender and education level of nursing staff is not meaningful and is unrelated to the research topic.

Answer: We give the data for statistical purposes only.

****************************************************************************************************************

The authors should specify the specific training content based on the results of this study in the training program.

Answer: We have done it, see the comments attached to the tables Page 8 ‐10 and the discussion

*************************************************************************************************************

Acknowledge any limitations of your study, such as the sample size or potential biases, to maintain transparency and reliability.

Answer: We have done it, lines 459 -463.

Round 2

Reviewer 1 Report

Comments and Suggestions for Authors

It is assessed as being appropriately corrected.

Author Response

Thank you for having accepted our work.

Reviewer 2 Report

Comments and Suggestions for Authors

The authors have made significant efforts to revise the manuscript, but there are still some issues that are recommended for further modification.

l   In journal writing, providing a clear statement on measures is essential. The authors should describe the rationale for developing the study variables, ensuring that the study variables and their measurement methods align with the concepts the research aims to investigate. The manuscript must provide details on the development of demographic and professional characteristics, the evolution of issues such as general knowledge, knowledge segregation practices, and training programs. Additionally, it should clarify the basis for subsequent inferential statistics, whether based on aggregation or averages.

l   The content of the questionnaire, measurement scales, and scoring methods should be described in the Materials and Methods section, not in the Discussion section.

l   In the revised manuscript, it is still unclear why six wards in N.G.H. were chosen.

l   The Statistical Analysis section must specify the statistical methods employed, such as t-test, ANOVA, or chi-square test.

l   The previous suggestion was to include the statistical mean values for each subcategory and the methods of analysis in Tables 3, 5, and 7. However, the revised manuscript does not reflect this change.

Author Response

Please find our comments to your responses below:

The authors have made significant efforts to revise the manuscript, but there are still some issues that are recommended for further modification.

Ans: Thank you for your positive comments. Below, we explain the modifications to our work according to your suggestions.

**********************************************************************************************************

l   In journal writing, providing a clear statement on measures is essential. The authors should describe the rationale for developing the study variables, ensuring that the study variables and their measurement methods align with the concepts the research aims to investigate. The manuscript must provide details on the development of demographic and professional characteristics, the evolution of issues such as general knowledge, knowledge segregation practices, and training programs. Additionally, it should clarify the basis for subsequent inferential statistics, whether based on aggregation or averages.

Ans: We provided our comments (and added new paragraphs) in sections 2.1, 2.2, 2.3.1, 2.3.2, 2.3.3, 2.3.4, 2.3.5 and 2.4.

*******************************************************************************************************

l   The content of the questionnaire, measurement scales, and scoring methods should be described in the Materials and Methods section, not in the Discussion section.

Ans: We have done so in sections 2.1, 2.2, 2.3.1, 2.3.2, 2.3.3, 2.3.4, 2.3.5 and 2.4.

*****************************************************************************************************

l   In the revised manuscript, it is still unclear why six wards in N.G.H. were chosen.

Ans: We provided an answer in the lines 135 – 137.

****************************************************************************************************

l   The Statistical Analysis section must specify the statistical methods employed, such as t-test, ANOVA, or chi-square test.

Ans: We changed the statistical sections according to your suggestions. Please see lines 205 -207

 ******************************************************************************************************

l   The previous suggestion was to include the statistical mean values for each subcategory and the methods of analysis in Tables 3, 5, and 7. However, the revised manuscript does not reflect this change.

Ans: We have modified Tables 3, 5, and 7 according to your suggestions by providing the statistical mean value of each subcategory and the type of each hypothesis testing according to your requests.

**********************************************************************************************************

Round 3

Reviewer 2 Report

Comments and Suggestions for Authors

Except for a minor suggestion below, the manuscript has been appropriately revised in all other aspects. With these corrections, the manuscript is now suitable for publication. Work experience at N.G.H. in Table 3 should be analyzed using ANOVA, not t-test. Additionally, ANOVA stands for Analysis of Variance, and each word should be capitalized.

Comments on the Quality of English Language

Minor editing of English language required

Author Response

I am attaching the final version with the corrections regarding the last comment.
